# The Urinary Exosomal miRNA Expression Profile is Predictive of Clinical Response in Lupus Nephritis

**DOI:** 10.3390/ijms21041372

**Published:** 2020-02-18

**Authors:** Eloi Garcia-Vives, Cristina Solé, Teresa Moliné, Marta Vidal, Irene Agraz, Josep Ordi-Ros, Josefina Cortés-Hernández

**Affiliations:** 1Vall d’Hebron University Hospital, Vall d’Hebron Research Insitute (VHIR), Rheumatology Research Group, Lupus Unit, 08035 Barcelona, Spain; eloigarvi@gmail.com (E.G.-V.); fina.cortes@vhir.org (J.C.-H.); 2Vall d’Hebron University Hospital, Department of Internal Medicine, 08035 Barcelona, Spain; jordi@vhebron.net; 3Vall d’Hebron University Hospital, Department of Renal Pathology, 08035 Barcelona, Spain; teresa.moline@vhir.org; 4Parc Taulí Foundation-UAB University Institute, Department of Renal Pathology, 08208 Barcelona, Spain; mvidal@tauli.cat; 5Vall d’Hebron University Hospital, Department of Nephrology, 08035 Barcelona, Spain; iagraz@vhebron.net

**Keywords:** lupus nephritis, urinary exosomes, biomarkers, response, therapy

## Abstract

Data on exosomal-derived urinary miRNAs have identified several miRNAs associated with disease activity and fibrosis formation, but studies on prognosis are lacking. We conducted a qPCR array screening on urinary exosomes from 14 patients with biopsy-proven proliferative lupus glomerulonephritis with a renal outcome of clinical response (*n* = 7) and non-response (*n* = 7) following therapy. Validation studies were performed by qRT-PCR in a new lupus nephritis (LN) cohort (responders = 22 and non-responders = 21). Responder patients expressed significantly increased levels of miR-31, miR-107, and miR-135b-5p in urine and renal tissue compared to non-responders. MiR-135b exhibited the best predictive value to discriminate responder patients (area under the curve = 0.783). In vitro studies showed exosome-derived miR-31, miR-107, and miR-135b-5p expression to be mainly produced by tubular renal cells stimulated with inflammatory cytokines (e.g IL1, TNFα, IFNα and IL6). Uptake of urinary exosomes from responders by mesangial cells was superior compared to that from non-responders (90% vs. 50%, *p* < 0.0001). *HIF1A* was identified as a potential common target, and low protein levels were found in non-responder renal biopsies. *HIF1A* inhibition reduced mesangial proliferation and *IL-8*, *CCL2*, *CCL3*, and *CXCL*1 mesangial cell production and *IL-6/VCAM-1* in endothelial cells. Urinary exosomal miR-135b-5p, miR-107, and miR-31 are promising novel markers for clinical outcomes, regulating LN renal recovery by *HIF1A* inhibition.

## 1. Introduction

Lupus nephritis (LN), occurring in ~40%–75% of patients with systemic lupus erythematosus (SLE), is one of the most severe forms of the disease, with an unpredictable course. Despite modern therapeutic approaches, LN is still a major cause of short and long-term morbidity [1], with up to 20% of patients progressing to end-stage renal disease (ESRD) [2,3]. Prompt diagnosis, treatment, and attainment of complete remission at early stages are clearly associated with better prognoses. 

So far, renal biopsy continues to be the gold standard for diagnosing and classifying scarring and the degree of renal inflammation, but its invasiveness is one of the main limitations for serial monitoring. To date, routine clinical parameters are not sensitive or specific enough for detecting ongoing disease activity and progression, early relapse, or response to therapy [4]. In recent years, several novel biomarkers have been identified to predict disease activity, but not many have been rigorously validated in large-scale longitudinal studies [5]. 

Extracellular vesicles (EVs) are a heterogeneous group of particles defined by size, composition, site of origin, and density, released by almost any cell, and include exosomes, microvesicles, and apoptotic bodies. During their formation, they incorporate different bioactive molecules from their cell of origin, such as soluble proteins, membrane receptors, nucleic acids (mRNAs and miRNAs), and lipids, which in turn can be transferred to target cells [6]. 

The discovery of miRNAs in various biological fluids suggests that miRNAs may be functioning as paracrine or endocrine signals between cells [7]. The fact that exosomal-derived urinary miRNAs can accurately reflect structural damage and renal dysfunction makes them good biomarkers for the diagnosis and prognosis of renal diseases [8]. So far, several studies have identified distinct miRNA expression profiles of urinary exosomes in patients with focal segmental glomerulosclerosis (FSG), diabetic nephropathy, and idiopathic nephrotic syndrome [9,10,11]. In LN, several exosomal-derived miRNAs have been identified to be markers of early fibrosis [12,13], podocyte injury [14], type IV class of nephritis [15], and the presence of cellular crescents [16], and can discriminate active LN [17]. To date, not much is known of their role as predictors of response to therapy. 

The aim of the study was to identify a non-invasive miRNA profile predictive of clinical response in LN and determinate their role in renal recovery. 

## 2. Results

### 2.1. Differentially Expressed Urinary Exosomal miRNA Profile in Patients with LN According to Clinical Renal Outcome

Fourteen patients with biopsy-proven proliferative lupus glomerulonephritis with a renal outcome of clinical response (*n* = 7) and non-response (*n* = 7) following conventional therapy were studied (Table 1). Urinary exosomes were isolated from urine collected during renal flare and 12 months post-treatment. They revealed spherical structures of 93.4 ± 36.6 nm diameter with the characteristic cup-shaped exosome morphology, and the RNA extracted from them showed an enrichment of small RNA species without ribosomal RNA presence (Appendix A). 

Four comparative analyses to identify differentially expressed (DE) miRNAs between responders and non-responders were carried out (Appendix A). Using a volcano plot, we identified 15 miRNAs with |log2fold change| > 10 and *p* < 0.05 (Figure 1A), and they were selected for the validation phase (Appendix A). To validate them, a new cohort of LN patients (responders *n* = 22 and non-responders *n* = 21, Table 2) was added to the initial screening cohort. 

Significantly increased miR-31-5p and miR-107 expression levels in the responder group were validated between flare and post-treatment pair-matched analysis (2.68- and 2.74-fold change, respectively, Figure 1B). Comparison at flare and post-treatment between responders and non-responders demonstrated an upregulation of miR-31-5p, miR-107, and miR-135b-5p levels in the responder group (Figure 1C). 

MiR-135b-5p correlated inversely with the degree of proteinuria in both groups, but the correlation was strongest in the non-responder group (r = −0.610, *p* = 0.008, Figure 1D). Histologically, only miR-31-5p was found to be inversely correlated with the renal activity score in the responder (r = −0.463, *p* = 0.026, Figure 1D). We found miR-135b-5p to have the best receiver operator characteristic (ROC) profile to distinguish responders from non-responders at flare time (area under the curve, AUC = 0.783, 77.8% sensitivity and 71.4% specificity, Figure 2A) and post-treatment (AUC = 0.855, 81% sensitivity and 79% specificity, Figure 2B). 

### 2.2. In Situ Renal Tissue Hybridization 

Renal tissue samples from responder patients showed an increased upregulation of miR-31-5p, miR-107 and miR-135b-5p compared with non-responders (2.27, 13.3 and 3.79-fold change fluorescence intensity, respectively, Figure 3A). All miRNAs were predominantly localized in the tubular cells, with the exception of miR-135b-5p that also showed some degree of glomerular staining (Figure 3B). By hematoxylin–eosin staining, levels of tubular atrophy were similar between groups (Figure 3C). 

### 2.3. Urinary Exosome-Derived miR-31-5p, miR-107 and miR-135b-5p Are Secreted by Stimulated Tubular Cells and Internalized Mainly by Mesangial Cells 

To investigate in vitro those cells responsible for the production of the selected miRNAs, primary renal cells were stimulated by a combination of inflammatory cytokines (IL1α, IL1β, TNFα, IFNα and IL6). At different time points of stimulation, exosomes were harvested from the culture supernatant to analyze the expression levels of the study miRNAs (Figure 4A). Expression of miR-31-5p, miR-107, and miR-135b-5p increased progressively over time reaching a peak at 18 h (Appendix A). At this time point, stimulated tubular renal cells displayed the highest expression levels of exosome-derived miR-31-5p, miR-107, and miR-135b-5p when compared with endothelial or mesangial cells (1.58-, 2.73-, and 3.52-fold change over control, respectively, *p* < 0.0001, Figure 4B). 

To evaluate which cellular type is responsible of the exosome uptake, we labelled purified urinary exosomes from responder and non-responder patients and co-incubated them in vitro with mesangial, tubular and endothelial primary cells (Figure 4C). After 3 h of co-incubation, uptake of urinary exosomes from responder patients by mesangial and endothelial cells was higher in comparison to the one observed from non-responder patients (55% vs. 5% and 85% vs. 15% fluorescence intensity, *p* = 0.002 and 0.0001, respectively, Figure 4C). At 18 h, uptake by endothelial cells was similar in both groups. However, non-responder exosomes uptake by mesangial cells was only of ~50% compared to ~90% of the urinary exosomes from responders (*p* < 0.001, Figure 3C). Exosome uptake by tubular cells was high (~80%) at all time-points, but no differences were observed between exosomes of different origin (Figure 4C). 

### 2.4. Renal HIF1A Expression is Reduced in Responder LN Patients via Exosomal miRNAs

We identified 177, 281, and 64 target genes for miR-31-5p, miR-107, and miR-135b-5p, respectively (Appendix A). Since the three miRNAs were overexpressed simultaneously in the responder group, we focused on identifying common targets. We identified six common target genes associated with renal diseases, and three (*HIF1A, FOXO1, KLF4*) are related specifically with LN (Figure 5A). Next, we measured protein levels of *HIF1A*, *FOXO1,* and *KLF4* in kidney biopsies from responder and non-responder patients. No differences in *FOXO1* and *KLF4* staining were observed in the study groups (Appendix A). However, *HIF1A* staining was higher in both glomerular and tubulointerstitial non-responder renal biopsies (average score of 1.8 and 2.2, respectively, Figure 5B). The difference in HIF1A protein levels between responders and non-responders was more significant in the glomerulus (*p* = 0.004, Figure 5B). 

To confirm whether *HIF1A* was a direct target of the studied miRNAs, we performed luciferase assay studies using primary human mesangial and endothelial cells. Luciferase activity decreased by 41.09%, 37.42%, and 30.37% in mesangial renal cells 48 h after the transfection in the presence of miR-135b-5p, mir-107, and miR-31-5p analogues, respectively, compared with the negative control (*p* < 0.001). In endothelial renal cells the luciferase activity decreased by 32.42%, 25.18%, and 18.80%, respectively (Figure 5C). 

### 2.5. Simultaneous Modulation of Candidate miRNAs Induces a Superior HIF1A Inhibition in Primary Renal Cells and in the Production of Inflammatory Molecules.

Since urinary exosomes were differentially uptake by mesangial and endothelial cells, we conducted the in vitro studies in those cellular types. Individual overexpression of each candidate microRNA led to a significant *HIF1A* downregulation following stimulation (*p* < 0.001, Appendix A). However, the simultaneous overexpression of miR-135-b/miR-107/miR-31 induced in both renal cell types a significant synergistic inhibitory effect (fold changes between −4.8 and −38.3, Figure 6A). *HIF1A* inhibition was also confirmed at a protein level by immunofluorescence (Appendix A).

Simultaneous overexpression of miR135-b/miR-107/miR-31 in mesangial cells led to a significant reduction of mesangial renal cell proliferation (*p* < 0.0001, Figure 6B) and pro-inflammatory cytokine production such as *IL6* and *IL8* (fold changes between −9.1 and −2.9, respectively, Figure 6C). In addition, a reduction in proinflammatory cytokines, *CXCL1*, *CCL2,* and *CCL3*, was observed following IL1α and INFα stimulation (*p* < 0.01, Figure 6C). In endothelial renal cells, the simultaneous overexpression of these miRNAs significantly inhibited the production of *IL6* following IL1α and VEGF stimulation (−7.7 and −4.2 fold change, respectively) and *VCAM-1* (−8.7 and −2.2 fold change, respectively, Figure 6D). 

### 2.6. Exosomes from LN Responder Patients Reduces the Production of Inflammatory Cytokines

In order to demonstrate that the influence of exosomes on mesangial and endothelial renal cells could induce renal recovery by *HIF1A* inhibition and downregulation of proinflammatory cytokines, we incubated exosomes from LN responder and non-responder patients with renal cells. A downregulation of *HIF1A* was observed with responder exosomes in both renal cells (fold change of −2.3 in mesangial and −4.1 in endothelial renal cells, Figure 7). In addition, a reduction of *CXCL1*, *CCL2*, *CCL3,* and *IL6* was also obtained. However, *IL8* was not significantly changed. In endothelial renal cells, a significant reduction of *IL6* and *VCAM-1* was observed (*p* < 0,05, Figure 7).

## 3. Discussion

In recent years there has been a growing interest in identifying exosome-derived miRNAs as biomarkers of LN. We performed a qPCR array screening comparing biopsy-proven LN patients who had responded to standard therapy with non-responders. Baseline variables were similar, except for a higher proportion of male patients in the non-responder group. Male gender, younger age, nephrotic syndrome and high chronicity index on renal biopsy are known poor prognostic factors in LN [18]. 

This is the first study to identify a specific urinary exosomal signature of miRNAs predictive of clinical response. Specifically, we found urinary miR-31-5p, miR-107, and miR-135b-5p levels to be highly expressed in responder patients during renal flare compared to non-responders; and in addition, levels remained higher for at least one-year post-treatment, suggesting a possible role for these miRNAs in renal recovery. ROC curve analysis showed that the study miRNAs had the capacity to discriminate between responders and non-responders, although miR-135b-5p showed the best predictive profile, with a sensitivity of 77.8% and a specificity of 71.4% to discriminate clinical response. Correlation analysis with clinical parameters found an association between urinary exosome miR-135b-5p expression levels and proteinuria; and miR-31 levels to be inversely correlated with renal activity score in the responder group. 

The hybridization in situ results mirrored the urinary miRNA expression levels, showing an increased expression of miR-31-5p, miR-107, and miR-135b-5p in the renal tissue of the responder group compared to non-responders. Interestingly, most miRNA expression was localized in the epithelial tubular cells, although miR-135b-5p was also mildly expressed in the glomeruli. The overexpression of these miRNAs was not associated with an increased level of tubular atrophy in the non-responder group. We performed in vitro studies to identify whether the tubular epithelial cells were the cell of origin of these miRNAs, or the target. Following exposure to labelled urinary exosomes from responder and non-responder patients, primary tubular epithelial cells exhibited a significant uptake (~80%) of both types of exosomes. However, cytokine-stimulated epithelial tubular cells significantly overexpressed these three miRNAs in comparison to the other cultured renal cells. These data suggest that differences observed in the hybridization in situ in the responder group are due to an increased production of the study miRNAs by tubular cells. The relevance of the epithelial cells in the secretion of exosomes into urine has been previously described. Exosomal signaling in the lumen of the renal nephron is unique. Since plasma exosomes cannot cross the glomerular filtration apparatus, intra-nephron exosomes originate exclusively from the luminal epithelial cells [19,20]. The study of potential recipient cells for these miRNAs showed a significantly higher uptake of responder urinary-labeled exosomes by endothelial cells at early stages, and mesangial cells over time. This data suggest that these two cells may act as targets for these miRNAs and may play a role in renal repair. 

The fact that the three identified miRNAs were increased simultaneously in LN responder group suggested that they may share common targets. Hypoxia-inducible factor-1 alpha (HIF-1α) has been reported as common target of them [21,22,23]. HIF-1α is an important player for the development of renal diseases but its role is controversial; whereas it has a protective role promoting cellular adaptation to hypoxia or angiogenesis, it can exacerbate fibrosis in tubular epithelial cells, promote in vivo glomerulosclerosis and mesangial renal proliferation, and contribute to glomerular injury [22,23,24]. Kidney biopsies from responder patients showed a significant reduction in HIF-1α expression, suggesting HIF-1α to play a relevant role in renal recovery. Since microRNAs may have a different effect depending on the expression of other microRNAs and their cross-talking, we hypothesized that the combined intervention of the three overexpressed candidate miRNAs might have the potential to be significantly more effective in suppressing HIF-1α and therefore enhancing its effect. We proved that the simultaneous overexpression of miR-31-5p, miR-107, and miR-135b-5p synergically enhanced *HIF1A* downregulation. In vitro studies suggest that *HIF1A* downregulation could contribute to renal recovery by inhibiting mesangial renal cell proliferation and downregulating the expression of mesangial inflammatory chemokines (*CXCL1*, *CCL3,* and *CCL2*) and interleukin-6 (*IL6*). In addition, through its effect on endothelial renal cells, it could contribute to renal recovery by reducing the expression of *IL6* and *VCAM-1* (Figure 8). 

Previous studies in LN have shown urinary levels of HIF-1α to be associated with histologic chronicity changes [25] and found to be highly expressed in glomerular and tubulo-interstitial biopsies [26]. It has been suggested that may promote mesangial cell growth through the induction of proliferation and inhibition of apoptosis. Besides the effect on cell proliferation, HIF-1α has also been suggested to have an impact on immune regulation. HIF-1α overexpression may contribute to LN exacerbation by enhancing B lymphocyte development [27], T lymphocyte differentiation [28,29], and innate immune responses [30]. 

In summary, we have demonstrated that urinary exosomal miR-135b-5p, miR-107, and miR-31-5p levels may be used as early markers of lupus nephritis outcome. Interestingly, their overexpression could ameliorate renal disease by inhibiting HIF-1α. For first time, we have identified relevant miRNAs involved in LN renal recovery that could contribute to develop new therapeutic approaches. 

## 4. Materials and Methods 

### 4.1. Patients and Samples

All participants had a biopsy-proven active proliferative LN. The written informed consent was provided prior inclusion (study approved by Vall d’Hebron Hospital Ethics Committee, PI15/02117, 20 April 2015). All patients fulfilled ≥ 4 criteria of the American College of rheumatology (ACR) for SLE classification [31,32,33]. Renal biopsies were categorized according to the International Society of Nephrology/Renal Pathology Society Classification (ISN/RPS) [34] and rated with respect to activity (AI) and chronicity (CI) by microscopy examination [35]. Urine and blood were collected during renal flare and 12 months after treatment. At this later point, patients were classified as responders or non-responders [36]. The screening cohort included 14 patients (7 responders and 7 non-responders) and the validation cohort included 43 patients (22 responders and 21 non-responders). More detailed information can be found in Appendix A.

### 4.2. Exosomal miRNA Extraction

Exosomes were isolated from urine using the miRCURY™ Exosomes Isolation Kit—Cells, urine and CSF (Exiqon, Woburn, MA, USA) and from serum using the exoRNeasy Serum/Plasma Kit (Qiagen, Hilden, Germany) following the manufacturer’s instructions. NanoSight, cryo-TEM, and Western blot analysis were used to characterize exosome isolation. RNA from exosomes was extracted following the instructions of miRCURY™ RNA Isolation Kit—Cell & Plant (Exiqon, Woburn, MA, USA) (detailed information in Appendix A). 

### 4.3. MicroRNA qPCR-RT Arrays and Individual Assays

Quantitative real-time PCR was performed using the manufacturer’s instructions for miRCURY LNA™ Universal RT microRNA PCR (Exiqon, Woburn, MA, USA). Here, 96-well pre-designed human urinary exosome plates (Exiqon, 87 LNA miRNA PCR primer sets, Appendix A) were used to analyze the microRNA profiling using ABI PRISM 7000. Data obtained from the Exiqon MicroRNA arrays were analyzed by the Bioinformatic Unit at Vall Hebron (detailed information in Appendix A). Data were deposited in the Gene Expression Omnibus (NCBI) with the number GSE140643. For validation, the specific miRCURY LNA primer set and ExiLENT SYBR Green master mix (Exiqon, Woburn, MA, USA, Appendix A) were used to quantify miRNA expression by ABI PRISM 7000 (detailed information in Supplementary Material). 

### 4.4. Double Fluorescent in situ hybridization (ISH) Detection and Immunofluorescence in Renal Biopsies

This was performed using paraffin-embedded (FFPE) renal biopsies (*n* = 5 responder and *n* = 5 non-responders). Double fluorescent ISH assay was performed using two differently labeled probes and detected sequentially following the methodology described by Silahtaroglu [37]. Simultaneous hybridizations were performed by including the two probes (hsa-miR-31-FITC and miR-532-DIG or hsa-miR-135b-FITC and miR-107-DIG). For immunofluorescence, primary antibodies were incubated at 4 °C overnight following by secondary antibody incubation at room temperature for 2 h. More detailed information can be found in Appendix A.

### 4.5. Production of the Study-Derived Exosomes by Primary Renal Cells

Primary renal cell lines were cultured in defined medium to 70%–80% confluence and then further cultured (1 × 10^5^) for 2–3 days with bovine fetal serum depleted of exosomes (Exo-FBS, 5%, System Biosciences, Palo Alto, CA, USA). Cells were stimulated for 6, 18, and 36 h with a mixture of inflammatory cytokines described previously in in vitro LN studies [38,39]: IL1α (10 ng/µL, ThermoFisher, Waltham, MA, USA), IL1β (10 ng/µL, Abcam, Cambridge, UK), TNFα (10 ng/µL, ThermoFisher, Waltham, MA, USA), INFα (10 ng/µL, ThermoFisher, Waltham, MA, USA), and IL6 (10 ng/µL, Abcam, Cambridge, UK). The control was stimulated with sterile PBS. Supernatants obtained from cell culture were used for exosome purification following the manufacturer’s instructions for the miRCURY™ Exosomes Isolation Kit—Cells, urine and CSF (Exiqon, Woburn, MA, USA) and subsequent gene expression analysis by qPCR-RT. 

### 4.6. Fluorescently Labelled Urinary Exosome Internalization by Primary Renal Cells

Urinary exosomes from responder, non-responder, and healthy controls were extracted as described previously (*n* = 5 for each group). To fluorescently label, 100 µg of exosomes were incubated for 30 min at room temperature with labeling dye (465/635 nm excitation/emission) provided for the ExoGlow-Membrane EV Labeling kit (System Biosciences, Palo Alto, CA, USA). The unlabeled dye was removed using the PD SpinTrap G-25 buffer exchange column (GE Healthcare, Chicago, IL, USA). Afterwards, 10 µg of labelled urinary exosomes were added to 24-well plates with mesangial, endothelial, or epithelial tubular cells. Cell internalization was analyzed at 3, 6, and 18 h by immunofluorescence microscopy.

### 4.7. Target Identification for Studied miRNAs

MiR-31-5p, miR-107, and miR-135b-5p may regulate biological pathways by targeting multiple pathway-specific mRNAs. For each miRNA, we retrieved validated targets from three major miRNA-target datasets using the multiMiR R analysis package: miRecords, miRTarBase, and miRWalk. Since the three miRNAs were overexpressed simultaneously in the responder patient group, we listed all validated targets from each miRNAs and we crossed them to identify the common targets. From them, we identified three related with LN pathogenesis (*HIF1A, FOXO1, KLF4*). 

### 4.8. Luciferase Reporter Assay 

Mesangial renal cells or endothelial renal cells were co-transfected with the vector pEZX-MT01-HIF1A 3′UTR and mimic miR-107, miR-31, or miR-135b-5p using DharmaFECT Duo transfect reagent (Thermo Fisher Scientific, Waltham, MA, USA) according to manufacturer protocols. After 48 h, luciferase activity was measured using a Dual-Luciferase Reporter Assay System (Promega, Madison, WI, USA, detailed information in Appendix A). 

### 4.9. MiR-135/miR-107/miR-31 Transfection in Primary Renal Cells 

Lipofectamine reagents RNAiMAX (Invitrogen, Carlsbad, CA, USA) were used to transfect mimics or anti miRNAs (Thermo Fisher, Waltham, MA, USA) into the primary renal cells, according to the manufacturer’s protocol. After transfection, cells were stimulated during 6 h, followed by total RNA extraction using miRCURY RNA Isolation Kit (Exiqon, Waltham, MA, USA). RNA concentration was obtained by Nanodrop. RNA was kept at −80 °C until use. 

### 4.10. Gene Expression Analysis by Quantitative Reverse Transcription Polymerase Chain Reaction (qPCR-RT)

For qPCR-RT mRNA, cDNA was obtained using RNA-to-cDNA Kit, and the quantificative RT-PCR reaction was performed using Taqman gene expression assay (Applied Biosystems, Foster City, CA, USA, Appendix A) on the ABI PRISM 7000HT. To choose endogenous genes, we evaluated four candidates (*GADPH*, *18S*, *PGK1* and *ACTB*). For our experiments, we used *GADPH* and *18S* as endogenous genes because they were the steadiest genes. Relative gene expression was obtained using Delta-Ct methods following the MIQE guidelines [40].

### 4.11. Patient’s Exosome Incubation into Primary Renal Cells

A mix of purified exosomes from three responders, non-responders, or healthy controls were incubated into mesangial and endothelial renal cells during 24 or 6 h, respectively. Incubation times were chosen according exosome’s internalization results, when they have higher internalization. After that, cells were lysed to obtain their RNA to analyze interested gene by qPCR-RT, as described previously. 

### 4.12. Statistical Analysis

GraphPad Prism (version 6, La Jolla, CA, USA) was used to perform the statistical analyses. MiRNA level expression was compared using the Mann–Whitney U test (non-parametric) and Student’s *t*-test or two-way ANOVA test (parametric) according to data distribution and number of groups. The correlation between two parameters was analyzed using Spearman’s rank-order correlation. ROC curves were used to evaluate the predictive value, sensitivity, and specificity of each biomarker, choosing the appropriate cutoffs according to Youden’s index. Statistical significance was set at *p*-value < 0.05.

## Figures and Tables

**Figure 1 ijms-21-01372-f001:**
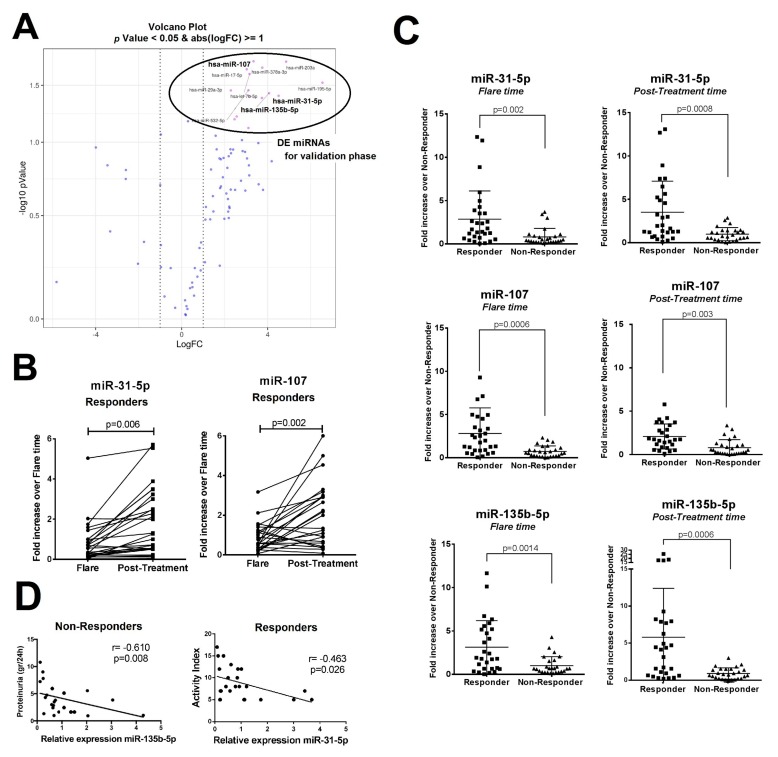
Urinary exosomal miRNA expression profile from lupus nephritis (LN) patients according to their clinical response. Expression of differentially expressed miRNAs was evaluated using quantitative real-time PCR (qRT-PCR) in the screening cohort and in the validation cohort of lupus nephritis patients (responders (*n* = 29) and non-responders (*n* = 28)). (**A**) Volcano plot shows the most differentially and significantly expressed miRNAs found in the screening cohort. Three of these mRNAs were validated (marked in bold). DE: differentially expressed. (**B**) Paired *t*-test analysis was done between flare and post-treatment time points in the responder group. Significant differences are shown with the corresponding *p*-values. (**C**) MiR-31-5p, miR-107, and miR-135b-5p expression from the LN responder and non-responder groups at flare time or post-treatment time are shown as individual plots. Gene expression was normalized using U6 as endogenous control. Fold change in expression level was calculated using the 2^−ΔΔCt^ method. *p*-Values were obtained using Student’s *t*-test. (**D**) Correlation of corresponding miRNAs expression levels with proteinuria and activity index at flare time. Spearman’s rank-order correlation was used to obtain *r-* and *p*-values.

**Figure 2 ijms-21-01372-f002:**
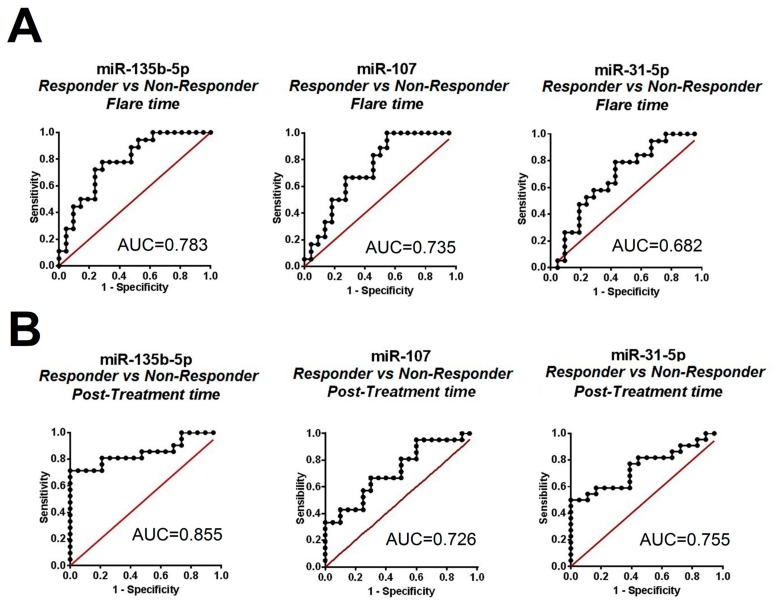
Differentially expressed microRNAs (miRNAs) according to treatment response. Receiver operator characteristic analysis (ROC) of individual miRNAs to distinguish responder from non-responder patient groups. Analysis from samples obtained at flare time (**A**) and at post-treatment time (**B**). The value of the area under the curve (AUC) is shown in each plot.

**Figure 3 ijms-21-01372-f003:**
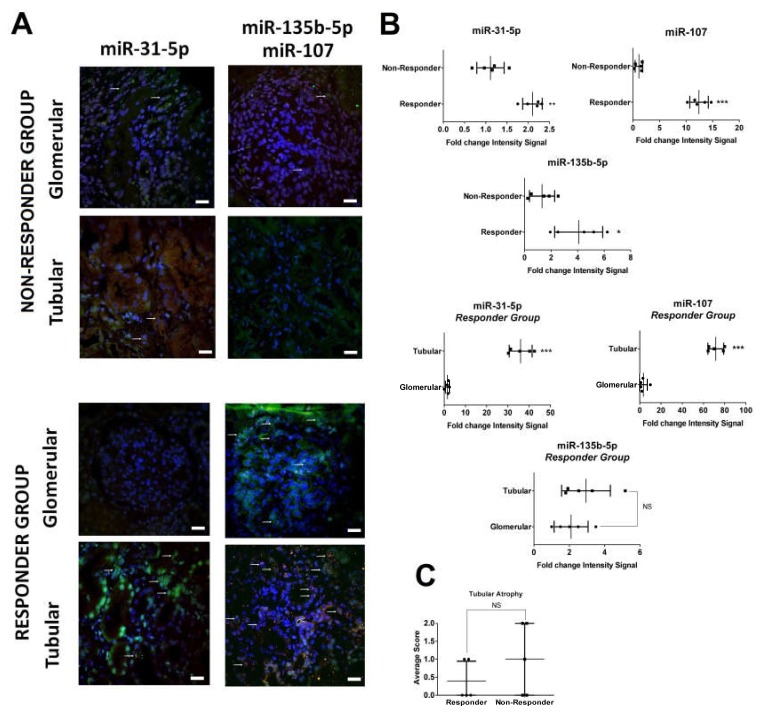
In situ hybridization of miR-31-5p, miR-135b-5p, and miR-107 in LN renal biopsies. (**A**) At flare time, renal tissue from lupus nephritis was analyzed to identify miR-31-5p, miR-135b-5p (both in green), and miR-107 (red). Significant differences were found between the responder and non-responder groups. (**B**) Tubular localization was found for miR-31 and miR-107 in the responder group. (**C**) No differences in tubular atrophy were observed between the responder and non-responder groups in histology analysis. Scale bar = 20 µm. White arrows marked positive cells. *p*-Values were obtained using Student’s *t*-test. NS: not significant, * *p* < 0.05; ** *p* < 0.005; *** *p* < 0.0005.

**Figure 4 ijms-21-01372-f004:**
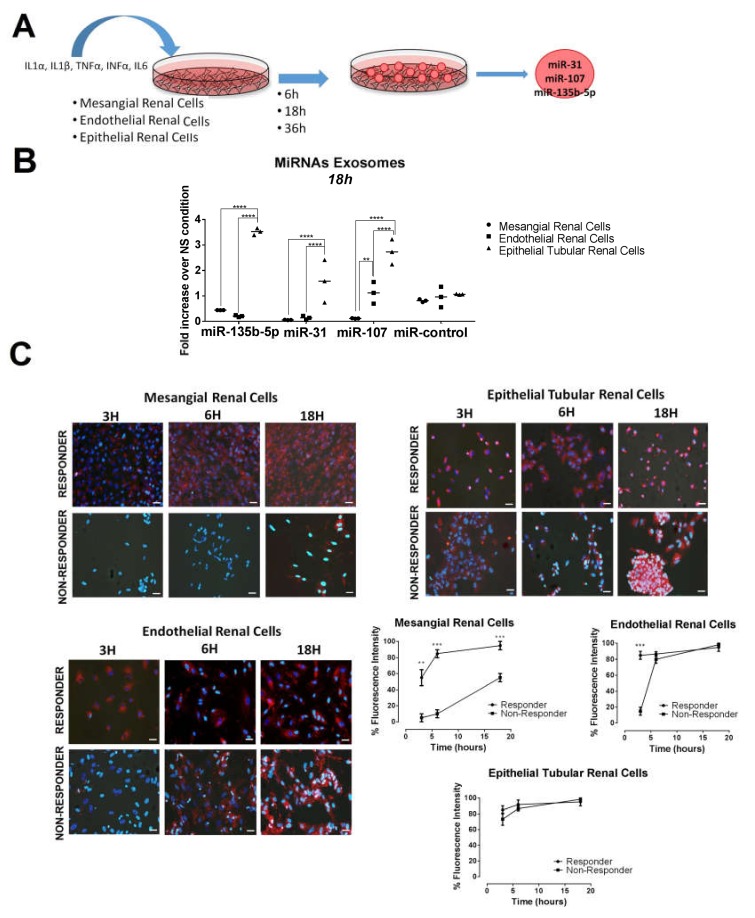
MiR-31-5p, miR-107, and miR-135b-5p urinary exosome origin and their target cell delivery. (**A**) Scheme of in vitro experiments performed to evaluate miRNA exosome production in mesangial, endothelial, and epithelial tubular renal cells. (**B**) After 18 h of stimulation, exosomes were obtained from studied renal cells to quantify their miRNA expression levels. Individual plots from *n* = 3 experiments. MiR-control is U6 small nuclear RNA. (**C**) Exosome internalization was measured by immunofluorescence in mesangial, endothelial, and epithelial tubular cells at different times. Blue color (DAPI) labels cell nuclei and red color labels exosomes. Each condition was assayed in triplicate. Error bars represent the means ± SEM from three independent experiments. Scale bar = 20 µm. *p*-Values were obtained using two-way ANOVA test. ** *p* < 0.005; *** *p* < 0.0005; **** *p* < 0.0001.

**Figure 5 ijms-21-01372-f005:**
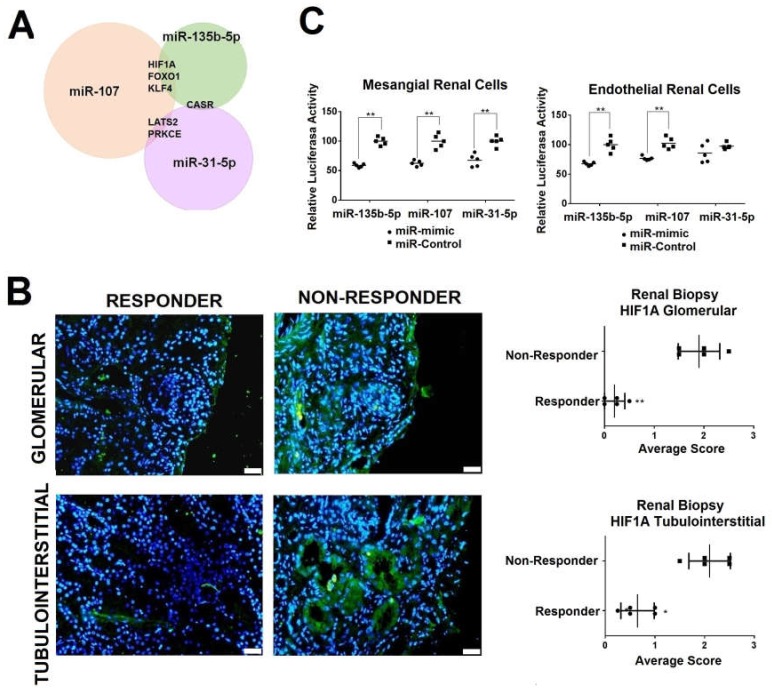
Urinary exosomal miRNAs specific for the LN responder group with their target mRNAs. (***A***) Venn diagram representing overlap of validated targets of miR-31-5p, miR-107, and miR-135b-5p. (**B**) Hypoxia-inducible factor-1 alpha (HIF1A) protein levels (green) in renal kidney biopsies from responder and non-responder patients. DAPI staining was used to label cell nuclei. Scale bar = 50 µm. * *p* < 0.05; ** *p* < 0.005. (**C**) Luciferase assay showed *HIF1A* as common target gene for miR-31, miR-107, and miR-135b-5p in mesangial or endothelial primary renal cells (five replicates per group). *p*-Values were obtained using two-way ANOVA test. ** *p* < 0.005.

**Figure 6 ijms-21-01372-f006:**
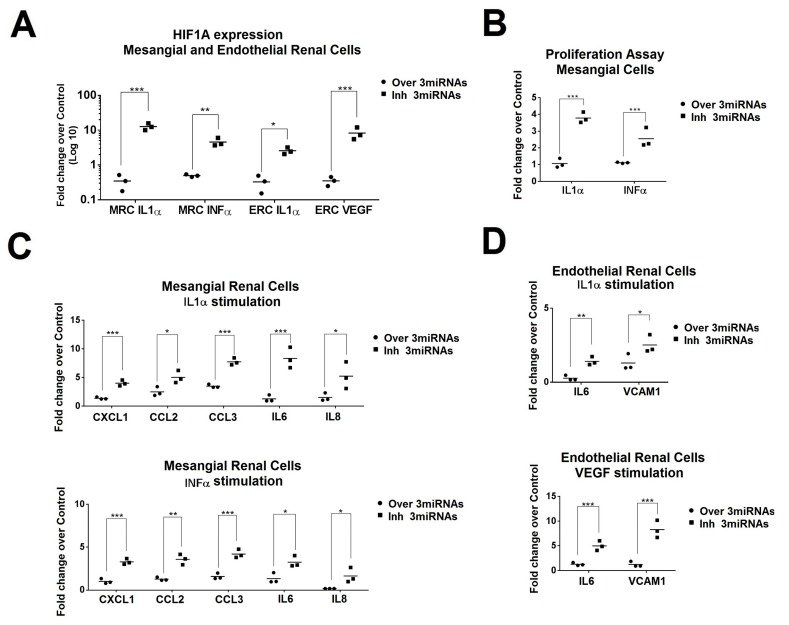
Effect of exosomal miRNAs to mesangial and endothelial renal cells. (**A**) *HIF1A* expression levels were analyzed by quantitative real-time in overexpressed or inhibited primary mesangial and endothelial renal cells after interleukin-1 alpha (IL1α), interferon alpha (INFα) or vascular endothelial growth factor (VEGF) stimulation. Overexpression or inhibition was performed for the three studied miRNAs (miR-31-5p, miR-107, and miR-135b-5p). MRC = mesangial renal cells. ERC = endothelial renal cells. Values were normalized using *GADPH* and *18S*. Data are expressed in Log10 scale. *** *p* < 0.0005, ** *p* < 0.005, and * *p* < 0.05. (**B**) Fold change of proliferation in mesangial renal cells with overexpression or inhibition of the miR-31, miR-107, and miR-135b-5p (over 3miRNAs or Inh 3miRNAS, respectively). Control conditions were obtained with mimic or anti miR-control. *** *p* < 0.0005. (**C**) Quantitative real-time RT-PCR analysis shows the relative mRNA levels of *CXCL1*, *CCL2, CCL3, IL6,* and *IL8* in IL1α- and INFα-stimulated renal mesangial cells (MRCs). Values were normalized using *GADPH* and *18S*. Fold change was calculated over the control condition (mimic miR-control or anti miR-control). *** *p* < 0.0005, ** *p* < 0.005 and * *p* < 0.05. (**D**) Expression levels of *IL6* and *VCAM-1* in endothelial renal cells with overexpression or inhibition of miR-31, miR-107, and miR-135b-5p after IL1α and VEGF stimulation. Fold change in expression level was calculated using the 2^−ΔΔCt^ method and over control condition (mimic miR-control or anti miR-control). *p*-Values were obtained using two-way ANOVA test. *** *p* < 0.0005, ** *p* < 0.005 and * *p* < 0.05.

**Figure 7 ijms-21-01372-f007:**
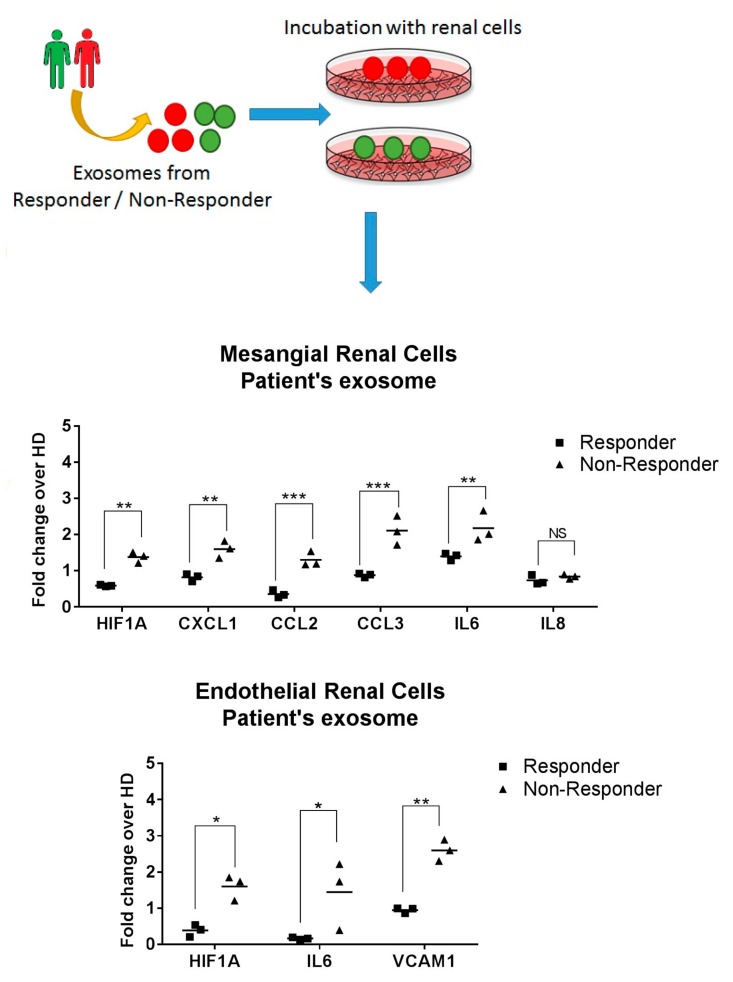
Role of patient exosome in renal recovery. Exosomes from LN responder or non-responder patients were incubated with mesangial and endothelial renal cells. After that, gene expression analysis shows a significant downregulation of *CXCL1, CCL2, CCL3, IL6,* and *VCAM-1.* Values were normalized using glyceraldehyde 3-phosphate dehydrogenase (*GADPH)* and 18S ribosomal RNA (*18S rRNA)*. Fold change was calculated over exosomes obtained from healthy donors (HD). *p*-Values were obtained using two-way ANOVA test. NS: not significant, *** *p* < 0.0005, ** *p* < 0.005 and * *p* < 0.05.

**Figure 8 ijms-21-01372-f008:**
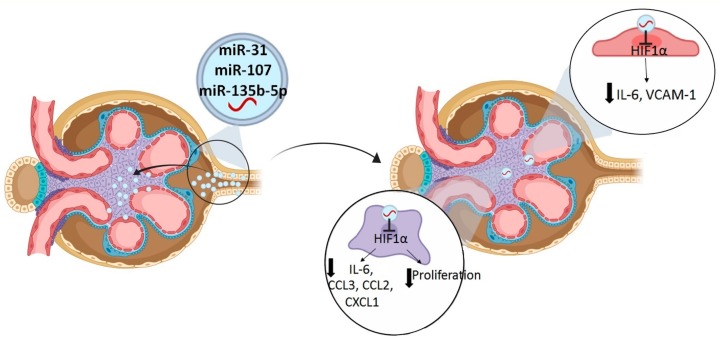
Proposed mechanism for urinary exosomal miRNAs in lupus nephritis renal recovery. Urinary exosomes from LN responder patients with high levels of miR-31-5p, miR-107, and miR-135b-5p are produced mainly in tubular renal cells to be engulfed by endothelial and mesangial renal cells. MiRNAs target HIF1A in endothelial renal cells, reducing the production of *IL6* and *VCAM-1*. In mesangial renal cells, miRNAs target *HIF1A*, inhibiting mesangial renal cell proliferation and downregulating the production of *CXCL1, CCL3, CCL2,* and *IL6*.

**Table 1 ijms-21-01372-t001:** Baseline demographic, laboratory, and clinical data from the screening cohort.

	Responders (*n* = 7)	Non-Responders (*n* = 7)	*p*-Value
**Demographic**			
Age (years), median (IQR)	36.9 (22.5–41.2)	27.9 (22.9–37.5)	0.330
Duration of SLE (years), median (IQR)	3.1 (0.1–13.3)	7.2 (5.9–9.2)	0.981
Sex, male/female	2/5	6/1	0.031
**Race/ethnicity, n (%)**			
Caucasian	6 (85.7%)	5 (72.1%)	0.700
Hispanic	1 (14.3%)	2 (28.6%)	0.700
**Laboratory parameters, median (IQR)**			
Serum creatinine, mg/dL	1.1 (1.0–1.3)	1.1 (0.7–1.5)	0.551
eGFR (mL/min/1.73 m^2^)	66 (48–98)	65 (30–101)	0.901
Anti-dsDNA Abs, IU/mL	79 (14–292)	73 (46–580)	0.372
Serum C3, mg/dL	61 (24–85)	69 (53–90)	0.309
Serum C4, mg/dL Hemoglobin, g/dL Leucocytes, ×10^−9^/L Lymphocytes, ×10^−9^/L ESR, mm/h Protein, g/dL Albumin, g/dL	8 (7–11) 11.6 (10.4–12.8) 5.7 (5.4–11.0) 1.3 (0.8–2.0) 33 (17–78) 6.5 (5.5–7.3) 3.3 (2.9–3.9)	11 (6–16) 11.6 (9.1–14.5) 7.4 (4.9–9.3) 0.9 (0.7–1.6) 51 (11–89) 5.1 (4.7–6.3) 2.8 (2.3–3.3)	0.635 0.983 0.893 0.208 0.746 0.114 0.100
Proteinuria, g/24 h Leucocituria Hematuria Cast	4.9 (1.4–6.8) 62 (29–241) 85 (17–300) 1 (1–7)	3.5 (1.6–6.5) 85 (30–174) 148 (42–470) 2 (0–15)	0.756 0.672 0.496 0.609
**Disease Activity, median (IQR)**			
SLEDAI-2K global score	20 (14–23)	18 (7–26)	0.409
Complete remission, *n* (%)	7 (100)	0 (0)	0.409
Time to remission (days), mean (SEM)	149 (25)	n.a.	n.a.
**Renal Biopsy, n (%)**			
Class III	2 (28.6%)	3 (42.9%)	0.999
Class IV	5 (71.4%)	4 (57.1%)	0.999
**Activity Index, median (IQR)**	10 (8–12)	7 (5–12)	0.607
Glomerular endocapillary proliferation	2 (1–2)	1 (1–2)	0.999
Glomerular neutrophilic infiltration	1 (1–2)	1 (0–2)	0.662
Wire-loop deposits and hyaline thrombi	1 (0–2)	1 (1–1)	0.999
Glomerular fibrinoid necrosis and karyorrhexis	2 (1–4)	2 (1–3)	0.858
Cellular crescents	2 (1–2)	2 (2–4)	0.312
Interstitial inflammation	1 (0–1)	1 (1–1)	0.442
**Chronicity Index, median (IQR)**	2 (0–4)	3 (1–4)	0.524
Glomerular sclerosis	0 (0–2)	1 (0–1)	0.877
Fibrous crescents	0 (0–1)	0 (0–1)	0.867
Tubular atrophy	0 (0–1)	0 (0–2)	0.505
Interstitial fibrosis	0 (0–1)	1 (0–1)	0.218

Values are expressed as median ± interquartile range (IQR). Title are in bold. SLE: systemic lupus erythematosus; ESR: erythrocyte sedimentation rate; eGFR: estimated glomerular filtration rate; anti-dsDNA: anti-double-stranded DNA (reference range <15 UI/mL); n.a: not applicable; SLEDAI-2K: systemic lupus erythematosus disease activity index 2000.

**Table 2 ijms-21-01372-t002:** Baseline demographic, laboratory, and clinical data according to clinical response from the second cohort.

Characteristics	Responders (n = 22)	Non-Responders (n = 21)	*p* Value
**Demographic**			
Age (years), mean	34.3 (2.3)	34.3 (2.3)	0.981
Duration of SLE (years), mean	7.1 (1.6)	5.8 (1.6)	0.571
Sex, male/female	2/20	8/13	0.034
Race/ethnicity, n (%)			
Caucasian	21 (95.5%)	18 (85.7%)	0.630
Hispanic	1 (4.5%)	3 (14.3%)	0.630
**Laboratory parameters, mean**			
Serum creatinine, mg/dL	1.10 (0.10)	1.3 (0.2)	0.387
eGFR (mL/min/1.73 m^2^)	73 (10)	68 (12)	0.710
Anti-dsDNA Abs, IU/mL	234 (54)	255 (68)	0.812
Serum C3, mg/dL	60 (6)	68 (4)	0.265
Serum C4, mg/dL Hemoglobin, g/dL Leucocytes, ×10^−9^/L Lymphocytes, ×10^−9^/L ESR, mm/h Protein, g/dL Albumin, g/dL	9 (1) 10.8 (0.5) 6.8 (0.6) 1.7 (0.4) 63 (7) 6.3 (0.4) 3.2 (0.2)	11 (1.4) 11.2 (0.5) 7.1 (0.8) 1.5 (0.3) 47 (8) 6.5 (0.3) 3.1 (0.2)	0.274 0.515 0.787 0.072 0.140 0.072 0.108
Proteinuria, g/24 h Leucocituria Hematuria Cast	3590 (736) 173 (49) 553 (286) 3 (2)	4589 (671) 149 (55) 308 (164) 4 (1)	0.322 0.750 0.472 0.416
**Disease Activity, mean**			
SLEDAI-2K global score	17.5 (2.5)	16.5 (1.5)	0.122
Complete remission, *n* (%)	22 (100)	0 (0)	0.122
Time to remission (days), mean (SEM)	214 (48)	n.a.	n.a.
**Renal Biopsy, n (%)**			
Class III	7 (31.8%)	8 (38.8%)	0.999
Class IV	15 (68.2%)	13 (61.2%)	0.999
**Activity Index, mean**	8.6 (1.1)	7.6 (1.2)	0.625
Glomerular endocapillary proliferation	1.6 (0.2)	1.6 (0.3)	0.913
Glomerular neutrophilic infiltration	1.0 (0.2)	0.9 (0.3)	0.849
Wire-loop deposits and hyaline thrombi	0.8 (0.4)	1.0 (0.2)	0.715
Glomerular fibrinoid necrosis and karyorrhexis	1.8 (0.6)	1.6 (0.5)	0.792
Cellular crescents	1.8 (0.5)	1.8 (0.5)	0.991
Interstitial inflammation	0.7 (0.3)	0.9 (0.2)	0.575
**Chronicity Index, mean**	1.8 (0.6)	2.4 (0.6)	0.487
Glomerular sclerosis	0.6 (0.3)	0.6 (0.2)	0.801
Fibrous crescents	0.2 (0.1)	0.2 (0.1)	0.700
Tubular atrophy	0.7 (0.2)	0.9 (0.2)	0.598
Interstitial fibrosis	0.6 (0.2)	0.8 (0.2)	0.594

Values are expressed as mean ± standard error median (SEM) or number of patients and percentage (*n*,%) when it was appropriated. SLE: systemic lupus erythematosus; ESR: erythrocyte sedimentation rate; eGFR: estimated glomerular filtration rate; anti-dsDNA: anti-double-stranded DNA (reference range < 15 UI/mL); n.a: not applicable. SLEDAI-2K: systemic lupus erythematosus disease activity index 2000.

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
