# Peer review of "The Urinary Exosomal miRNA Expression Profile is Predictive of Clinical Response in Lupus Nephritis"

_ijms, 2020, doi:10.3390/ijms21041372_

Round 1
Reviewer 1 Report
The paper by Garcia-Vives and colleagues reported a new interesting method to discriminate LN patients from responders to non responder at the pharmacological treatment.
I found the paper well written and exhaustive, and I think that there are only some points that need to be better clarified.
Major finding:
In the Methods section there is no description on how the Authors performed gene expression analysis, only the miRNA. In addition, while it is commonly accepted to perform miRNA qPCR analysis with the delta-delta-Ct method using only one stable miRNA, this is not true for gene expression analysis. In 2009 the guidelines on reporting qPCR gene expression analysis were published (PMID: 19246619) and these guidelines indicate the use of at least two endogenous genes in the data analysis while the authors indicated only GAPDH in the legend of figure 6. The authors should report gene expression data following these guidelines and add missing information.
Minor observations:
In the Abstract the Authors wrote "microarray screening" but they made "qPCR array screening" which are different technologies. This should be change.
In the overall text the Authors talk about stimulated cells, but I think that this should be better explained. First of all in the abstract section, and in the introduction they should add a couple of information on the cytokines used to explain why they choose these specific molecules rather than just writing "relevant to LN" in the results section and without adding references.
Table 1: since there are only 7 patients in each group, median and IQR should be used instead of mean and SD. Table 2: non all the variables are shown as mean and SD, so the legend would be completed with the missing information.
I noticed that the majority of the patients in both cohorts of non-responders are male (also confirmed by statistical analysis) while LN is usually more frequent in female patients. This should be an interesting point to be discuss.
The Authors did not described how they labeled purified exosomes from responders and non-responder patients. This information should be added.
Figure 4: in Figure legend there is no indication on what the colors are. miR-control should be described: is U6 or another one? I found nowhere the p<0.05 represented by one asterisk. Is the symbol missing in the figure or this is a non-necessary information?
Figure S2. The Authors did not described what “DE miRNA” means (maybe differentially expressed?). Please also confirmed that, in the heatmap, red means upregulated miRNAs. These information should be added in the figure legend.
Finally, please remember that when you talk about human genes they have to be written in capitol letters and italic.
Author Response
REVIEWER 1:
The paper by Garcia-Vives and colleagues reported a new interesting method to discriminate LN patients from responders to non responder at the pharmacological treatment.
I found the paper well written and exhaustive, and I think that there are only some points that need to be better clarified.
Major finding:
In the Methods section there is no description on how the Authors performed gene expression analysis, only the miRNA. In addition, while it is commonly accepted to perform miRNA qPCR analysis with the delta-delta-Ct method using only one stable miRNA, this is not true for gene expression analysis. In 2009 the guidelines on reporting qPCR gene expression analysis were published (PMID: 19246619) and these guidelines indicate the use of at least two endogenous genes in the data analysis while the authors indicated only GAPDH in the legend of figure 6. The authors should report gene expression data following these guidelines and add missing information.
We usually perform a screening of four endogenous genes (GAPDH, 18S, PGK1 and ACTB) when we start a new qPCR-RT analysis because GAPDH is not always the best reference gene. In this case, we evaluated them in our primary mesangial and endothelial renal cells in basal conditions as shown below:
We chose GADPH and 18S because they present lower Ct values, less differences between the two cell lines and less dispersion. Gene expression were normalized using GADPH and 18S as endogenous genes. Firstly, it was not specify into the manuscript but this information has been added and detailed this in the methodology (Page 17, lines 395-400) and we cited the manuscript “The MIQE guidelines: minimum information for publication of quantitative real-time PCR Experiments” that we used to follow for our gene expression analysis.
Minor observations:
In the Abstract the Authors wrote "microarray screening" but they made "qPCR array screening" which are different technologies. This should be change.
Changes had been made.
In the overall text the Authors talk about stimulated cells, but I think that this should be better explained. First of all in the abstract section, and in the introduction they should add a couple of information on the cytokines used to explain why they choose these specific molecules rather than just writing "relevant to LN" in the results section and without adding references.
To conduct the stimulation of primary renal cells we used different inflammatory cytokines such as IFNα, IL1, IL6 and TNFα. These cytokines have been previously shown to play a role in in vitro studies of LN [1,2]. Interferon-alpha (IFNα) is a key regulator of the innate immune system and IL-1 or IL-6 are pro-inflammatory cytokines relevant for the inflammatory response in lupus nephritis [1,2]. Tumor necrosis factor-alpha (TNF-α) is also a potent inflammatory mediator and apoptosis inducer that have been related with lupus nephritis pathogenesis [2]. Information about the cytokines used for in vitro experiments had been added in the abstract section and in methods (Page 16, lines 360-363).
[1] Lourenço EV, La Cava A. Cytokines in systemic lupus erythematosus. Curr Mol Med. 2009, 9, 242-254.
[2] Iwata Y, Furuichi K, Kaneko S, Wada T. The role of cytokine in the lupus nephritis. 2011. J Biomed Biotechnol. 2011, 2011, 594809.
Table 1: since there are only 7 patients in each group, median and IQR should be used instead of mean and SD. Table 2: non all the variables are shown as mean and SD, so the legend would be completed with the missing information.
Tables have been corrected and missing information has been added.
I noticed that the majority of the patients in both cohorts of non-responders are male (also confirmed by statistical analysis) while LN is usually more frequent in female patients. This should be an interesting point to be discuss.
After comparing baseline characteristics between responders and non-responders, we observed a higher number of male patients in the no-responder group. Although SLE affects predominantly women, and therefore also LN, several authors have reported male gender as a poor prognosis factor in LN. This has been included in the discussion (Page 13, lines 246-249).
The Authors did not described how they labeled purified exosomes from responders and non-responder patients. This information should be added.
This information has been added in the methodology (Page 16, lines 367-374).
Figure 4: in Figure legend there is no indication on what the colors are. miR-control should be described: is U6 or another one? I found nowhere the p<0.05 represented by one asterisk. Is the symbol missing in the figure or this is a non-necessary information?
We have included information about the color: red color shows labeled exosomes and blue color shows nuclei cells (DAPI). MiR-control is U6 so this information has been added. It is not necessary to include the information about one asterisk in this figure, since this condition is non-existing. We have done the changes as suggested.
Figure S2. The Authors did not describe what “DE miRNA” means (maybe differentially expressed?). Please also confirmed that, in the heatmap, red means upregulated miRNAs. This information should be added in the figure legend.
We have described the meaning of DE miRNAs that is “differentially expressed miRNAs” (Legend Figure S2). We confirmed that red color in the heatmap is for upregulated miRNAs. This information has been added in Figure S2.
Finally, please remember that when you talk about human genes they have to be written in capitol letters and italic.
Changes have been done.

Reviewer 2 Report
In their interesting study Eloi Garcia-Vives et al. investigated the utility of urinary exosome-derived miRNA in prediction of treatment response in lupus nephritis (LN) patients (87 miRNA-array in the screening cohort [n=14], 15 targets were next validated in larger cohort [n=43]). They identified 3 miRNA during validation phase (miR-31-5p, miR-107, miR-135b-5p) which were upregulated in responder subgroup at the time of renal flare. The authors traced the origin of these exosome-derived miRNA to activated tubular renal cells. In the in vitro experiments they also identified HIF1A as a common target for the three miRNAs, and confirmed that their over-expression resulted in inhibition of inflammatory response in endothelial and mesangial cells. I don have any major critics regarding the manuscript content.
Minor concerns
Figure legends not clear. Please update descriptions to match figure plots. Currently, they do not exactly describe what is shown in the data plots. For example, Fig. 1: (A) which data are actually shown here (responder or not responder) as there is only 1 graph for each miRNA). (C) This is responder at flare, I guess. Fig. 2: (A) It does not show any statistics as written in description, this probably applies to upper 3 graphs from 2B. Please point exemplary miRNA-expressing cell with arrow or symbol. The author should also consider adding info on statistical test used when explaining symbols of P-values in descriptions. Discrepancy between screening and validation data. The Authors found 3 urinary-exosome derived miRNAs downregulated in non-responder LN validation cohort. This was not observed in screening cohort, despite similar detection method. This is likely due to large dispersion of data (at least in responder group). The two cohorts are quite similar in terms of clinical characteristics. In my opinion, for validation part, it would be better to combine the two groups (n=57). I think, the Authors should present selected datasets in the screening part (at least A1 vs B1) in the form of volcano plot, highlighting miRNAs chosen for validation and the three studied in detail. It will probably show some trend (i.e. toward higher expression of the three miRNAs in the responder group) that might somehow correspond to validation data. Diagnostic usefulness of exosome derived miRNAs. The Authors found urine-exosome miR-135b-5p to be useful biomarker in prediction of treatment response in LN (AUC=0.78). Implementation of exosome-derived miRNAs in diagnostics might not be easy due to rather complicated methodology, and potential confounding factors. Did authors check for differences in miR-135b-5p expression simply in the urine supernatant (relative or per urine volume)? This can make the potential test easier to use. Number of patients. Please check number of replicates throughout text (e.g. responder /non-responder numbers are different in Table 2 and Fig. 1 legend). Please provide methods (and database citations) for selection of mRNA targets. The Authors used 3 online tools to identify gens listed in suppl. Table S6. I assume these were predicted targets, but what was methodology to shrunk the enormous number of targets proposed. If possible, please change scale to log10 in some graphs with miR-mimic and anti-miR experiments (e.g. Fig. 5D). I know, the data is very convincing, but this can help to evaluate the degree of inhibition in the “Over” conditions. The authors identified epithelial tubular cells as a main source for exosomes containing the three miRNAs analyzed (at least miR-135b-5p and miR-31). They confirmed potent uptake of non-responder exosomes by both mesangial and endothelial cells, and showed functional changes in these cells upon miRNA over-expression. The conclusions (e.g. presented in Fig. 6) suggest that exosomes enriched with miRNA are released by epithelial cells and target mesangial cells, influencing their reponses. To prove this, the Authors should provide data on the influence of exosomes on mesangial cell function, e.g. by incubating mesangial cells with (1) urine derived exosomes, (2) epithelial tubular renal cell derived exosomes (“Over” or “Inh” 3miRNAs to show potential dose dependence), (3) mesangial cell derived exosomes as a control. If this is not possible the conclusions should be more restrained. I suggest using ITALIC for human gene symbols.Author Response
REVIEWER 2:
In their interesting study Eloi Garcia-Vives et al. investigated the utility of urinary exosome-derived miRNA in prediction of treatment response in lupus nephritis (LN) patients (87 miRNA-array in the screening cohort [n=14], 15 targets were next validated in larger cohort [n=43]). They identified 3 miRNA during validation phase (miR-31-5p, miR-107, miR-135b-5p) which were upregulated in responder subgroup at the time of renal flare. The authors traced the origin of these exosome-derived miRNA to activated tubular renal cells. In the in vitro experiments they also identified HIF1A as a common target for the three miRNAs, and confirmed that their over-expression resulted in inhibition of inflammatory response in endothelial and mesangial cells. I don have any major critics regarding the manuscript content.
Minor concerns:
Figure legends not clear. Please update descriptions to match figure plots. Currently, they do not exactly describe what is shown in the data plots. For example, Fig. 1: (A) which data are actually shown here (responder or not responder) as there is only 1 graph for each miRNA). (C) This is responder at flare, I guess. Fig. 2: (A) It does not show any statistics as written in description, this probably applies to upper 3 graphs from 2B. Please point exemplary miRNA-expressing cell with arrow or symbol. The author should also consider adding info on statistical test used when explaining symbols of P-values in descriptions.
We have revised and corrected figure legends as suggested. We have added white arrows to label miRNA-expressing cells in Figure 3. We have added information about statistical test in each figure and in the methodology (Page 17, lines 408-411).
Discrepancy between screening and validation data. The Authors found 3 urinary-exosome derived miRNAs downregulated in non-responder LN validation cohort. This was not observed in screening cohort, despite similar detection method. This is likely due to large dispersion of data (at least in responder group). The two cohorts are quite similar in terms of clinical characteristics. In my opinion, for validation part, it would be better to combine the two groups (n=57). I think, the Authors should present selected datasets in the screening part (at least A1 vs B1) in the form of volcano plot, highlighting miRNAs chosen for validation and the three studied in detail. It will probably show some trend (i.e. toward higher expression of the three miRNAs in the responder group) that might somehow correspond to validation data.
We have combined the two cohorts to obtain more significantly results (Figure 1B,C,D). We also included in the manuscript the volcano plot showing miRNAs chosen for validation (Figure 1A) and we marked in bold the three validated miRNAs.
Diagnostic usefulness of exosome derived miRNAs. The Authors found urine-exosome miR-135b-5p to be useful biomarker in prediction of treatment response in LN (AUC=0.78). Implementation of exosome-derived miRNAs in diagnostics might not be easy due to rather complicated methodology, and potential confounding factors. Did authors check for differences in miR-135b-5p expression simply in the urine supernatant (relative or per urine volume)? This can make the potential test easier to use.
We extracted RNA from 500µl of fresh urine and also from cellular pellet. Cellular pellet were obtained from 50mL fresh urine after centrifugation (3.900g during 30 min at 4ºC). However, we obtained Ct >38 for miR-135b-5p in both patients groups (responder and non-responders). We considered that they are not the best source to determinate miR-135b-5p expression because high levels of Cts could make misunderstanding.
Number of patients. Please check number of replicates throughout text (e.g. responder /non-responder numbers are different in Table 2 and Fig. 1 legend).
We have checked and corrected.
Please provide methods (and database citations) for selection of mRNA targets. The Authors used 3 online tools to identify gens listed in suppl. Table S6. I assume these were predicted targets, but what was methodology to shrunk the enormous number of targets proposed.
We only considered validated targets from three major miRNA-taget datasets, miRecords, miRTarBase and miRWalk using the multiMiR R analysis package. We identified 177 (miR-31-5p), 281 (miR-107) and 64 target genes (miR-135b-5p). From them, we were only focus into the six common targets for the three miRNAs (Venn’s diagram in Figure 5A) because they are overexpressed in responder group. We have added this information in the methodology (Page 16, lines 375-381).
If possible, please change scale to log10 in some graphs with miR-mimic and anti-miR experiments (e.g. Fig. 5D). I know, the data is very convincing, but this can help to evaluate the degree of inhibition in the “Over” conditions.
We have changed the graph scale as suggested (Fig 6A).
The authors identified epithelial tubular cells as a main source for exosomes containing the three miRNAs analyzed (at least miR-135b-5p and miR-31). They confirmed potent uptake of non-responder exosomes by both mesangial and endothelial cells, and showed functional changes in these cells upon miRNA over-expression. The conclusions (e.g. presented in Fig. 6) suggest that exosomes enriched with miRNA are released by epithelial cells and target mesangial cells, influencing their reponses. To prove this, the Authors should provide data on the influence of exosomes on mesangial cell function, e.g. by incubating mesangial cells with (1) urine derived exosomes, (2) epithelial tubular renal cell derived exosomes (“Over” or “Inh” 3miRNAs to show potential dose dependence), (3) mesangial cell derived exosomes as a control. If this is not possible the conclusions should be more restrained.
In order to demonstrate that exosomes from LN responder patients could induce renal recovery, we extracted them from urine and incubated with mesangial or endothelial renal cells. The incubation were 24 and 6 hours, respectively. Incubation time were chosen according results obtained in exosome’s internalization (Figure 4C). After that, we extracted RNA to study HIF1A and cytokine gene expression. We observed that in mesangial renal cells HIF1A, CXCL1, CCL2, CCL3 and IL6 were downregulated after incubation of responder’s exosome. In endothelial renal cells, we observed downregulation of HIF1A, IL6 and VCAM1. So we added this result in the manuscript (Fig 7, Page 12, line 228-235), we changed our hypothesis (Page 14, line 290-292) and Figure 8 according this results.
I suggest using ITALIC for human gene symbols.
Changes have been done.

Round 2
Reviewer 1 Report
I think that this paper is acceptable in its present form.